# Effects of S-Adenosylhomocysteine Hydrolase Downregulation on Wnt Signaling Pathway in SW480 Cells

**DOI:** 10.3390/ijms242216102

**Published:** 2023-11-08

**Authors:** Ivana Pavičić, Filip Rokić, Oliver Vugrek

**Affiliations:** Laboratory for Advanced Genomics, Divison of Molecular Medicine, Institute Ruđer Bošković, Bijenička Cesta 54, 10000 Zagreb, Croatia; ivana.pavicic@irb.hr (I.P.); frokic@irb.hr (F.R.)

**Keywords:** AHCY deficiency, S-adenosylhomocysteine (SAH), LEF1, metastasis, EMT, cancer cell phenotype, shRNA, next-generation sequencing

## Abstract

S-adenosylhomocysteine hydrolase (AHCY) deficiency results mainly in hypermethioninemia, developmental delay, and is potentially fatal. In order to shed new light on molecular aspects of AHCY deficiency, in particular any changes at transcriptome level, we enabled knockdown of AHCY expression in the colon cancer cell line SW480 to simulate the environment occurring in AHCY deficient individuals. The SW480 cell line is well known for elevated AHCY expression, and thereby represents a suitable model system, in particular as AHCY expression is regulated by MYC, which, on the other hand, is involved in Wnt signaling and the regulation of Wnt-related genes, such as the β-catenin co-transcription factor LEF1 (lymphoid enhancer-binding factor 1). We selected LEF1 as a potential target to investigate its association with S-adenosylhomocysteine hydrolase deficiency. This decision was prompted by our analysis of RNA-Seq data, which revealed significant changes in the expression of genes related to the Wnt signaling pathway and genes involved in processes responsible for epithelial-mesenchymal transition (EMT) and cell proliferation. Notably, LEF1 emerged as a common factor in these processes, showing increased expression both on mRNA and protein levels. Additionally, we show alterations in interconnected signaling pathways linked to LEF1, causing gene expression changes with broad effects on cell cycle regulation, tumor microenvironment, and implications to cell invasion and metastasis. In summary, we provide a new link between AHCY deficiency and LEF1 serving as a mediator of changes to the Wnt signaling pathway, thereby indicating potential connections of AHCY expression and cancer cell phenotype, as Wnt signaling is frequently associated with cancer development, including colorectal cancer (CRC).

## 1. Introduction

S-adenosylhomocysteine hydrolase (AHCY) is an enzyme that catalyzes the hydrolysis of S-adenosylhomocysteine (SAH) to produce adenosine (Ado) and homocysteine (Hyc) [1,2,3]. SAH is generated through transmethylation reactions of S-adenosylmethionine (SAM), which serves as a main methyl-group donor in most living organisms, and is involved in a variety of cellular processes, including DNA methylation, histone modification, and RNA processing. Also, SAH is a potent competitive inhibitor of methyltransferases [4]. Therefore, maintaining proper AHCY activity is crucial for regulating the cellular methylation potential, which is determined by the ratio of SAH to SAM metabolites [5,6]. The significance of AHCY in regulating the cellular methylation potential has been underscored by the discovery of AHCY deficiency in humans, a rare and potentially lethal multisystem disorder caused by allelic mutations in the AHCY gene that results in reduced AHCY enzymatic activity [1,7,8,9,10,11], with dramatically increased levels of metabolites SAH and SAM. Evidently, increased levels of SAH are associated with increased cell proliferation, migration, and invasion, possibly due to disruption of methylation status and altered expression of key genetic factors that control these vital cellular processes [12]. In a previous study, we showed that DNA hypermethylation seems to be a frequent but not constant feature associated with AHCY deficiency that affects different genomic regions to different degrees, which, on the other hand, could impair the regulation of gene expression, and cellular signaling pathways, respectively [13].

So far, AHCY deficiency has been linked to a range of metabolic disorders, including liver disease [14]. Also, AHCY downregulation contributes to tumorigenesis [15,16,17].

However, the pathogenic effects of AHCY deficiency at molecular level still appear elusive. Understanding the mechanisms that lead to aversive conditions and diseases as a result of AHCY deficiency is important for developing effective strategies for the prevention and treatment of these conditions. Thus, in this study we investigated the effects of AHCY downregulation on the colon cancer derived model cell line SW480.

To do so, we enabled short hairpin RNA (shRNA)-mediated gene silencing in SW480 cells in order to resemble AHCY deficiency at cellular level. The efficiency of generating AHCY-deficient environments was evaluated by determining the concentrations of crucial metabolites SAM and SAH in the AHCY-deficient SW480 cells. Subsequently, after establishing effective AHCY silencing, we performed RNA sequencing to analyze changes in gene expression levels in response to AHCY down-regulation. To contextualize the RNA-seq data, we deployed ingenuity pathway analysis (IPA) for predicting changes to signaling networks and related downstream effects, and to identify new targets or candidate biomarkers in the context of various biological systems. We also performed Western blotting to confirm data obtained from RNA sequencing, by assessing the protein levels of key components of signaling pathways of interest.

Besides the major role of AHCY in the regulation of the cellular methylation potential, we show that AHCY downregulation can impact a wide range of cellular functions. Our findings provide new insights into the molecular mechanisms underlying the effects of AHCY downregulation, in particular on lymphoid enhancer-binding factor 1 (LEF1), a member of the T-cell Factor (TCF)/LEF1 family of high-mobility group transcription factors, and a downstream mediator of the Wnt/β-catenin signaling pathway. The Wnt pathway is highly conserved and involved in various cellular processes, including embryonic development, tissue homeostasis, stem cell maintenance, and cell differentiation. This result bodes well with previous findings that implicate disorder of Wnt signaling in various diseases, including cancer and neurodegenerative diseases [18]. On the other hand, LEF1 is essential in stem cell maintenance and organ development, especially in its role in epithelial-mesenchymal transition (EMT) [19]. Thus, the investigation of this relationship may help elucidate the molecular mechanisms underlying the effect of AHCY on cancer cell behavior and may contribute to a better understanding of the pathogenesis of AHCY deficiency-related diseases.

## 2. Results

### 2.1. SAM/SAH Measurements

The levels of S-adenosylmethionine (SAM) and S-adenosylhomocysteine (SAH) were measured in AHCY-deficient cells to investigate the impact of AHCY deficiency on the metabolism of these compounds. Compared to the control cells, the AHCY-deficient cells exhibited a significant increase in the amount of SAH, with levels approximately two-fold higher (Figure 1). In the AHCY deficient cells, the concentration of SAH was found to be 4.6 ng/mL, whereas in the control cells, it was 2 ng/mL. This substantial elevation in SAH suggests impaired methylation capacity in the AHCY deficient cells, as SAH is an inhibitor of methyltransferase enzymes. Conversely, although slightly lower, the levels of SAM, the precursor of SAH and a key methyl donor, did not show a significant difference between the AHCY deficient cells and the control cells. The concentration of SAM in the AHCY deficient cells was 278 ng/mL, while in the control cells, it was 458 ng/mL.

### 2.2. RNA-Seq

mRNA sequencing of SW480 cells yielded an average of 65 million paired-end reads per sample, with a read length of 151 nucleotides. The reads were aligned to the reference genome (hg38 No Alts, with decoys) using DRAGEN RNA Pipeline (Version 3.9.5) available on BaseSpace Sequence Hub (Illumina Inc., San Diego, CA, USA), with an average mapping rate of 98.1% of total reads. RNA quantification was performed using the aforementioned BaseSpace tool, and differential expression analysis was performed using the DRAGEN Differential Expression tool (Illumina Inc., San Diego, CA, USA) that utilizes DeSeq2 algorithm to identify differential expresses genes between two conditions.

Overall, we identified approx. 15,000 expressed genes in SW480 cells, with 3627 genes showing significantly different expression levels between treated and control groups (adjusted *p*-value < 0.05 and fold change > 0.5). Of these differentially expressed genes, 1350 were upregulated and 2277 were downregulated in the treated group compared to the control group.

### 2.3. In-Depth Examination of the Most Significantly Altered Signalling Pathways

The IPA software package (https://analysis.ingenuity.com/pa/) was used to analyze the data generated by mRNA sequencing, indicating significant differences in gene expression in several biological processes and pathways. We found 523 genes related to cellular movement, 605 related to cellular development, and 600 genes related to cellular growth and proliferation to be differentially expressed, respectively (Table 1 and Table 2).

Furthermore, we have focused our analysis to most relevant non-canonical signaling pathways that are somehow connected to differentially expressed LEF1 protein, such as the Wnt signaling network (Figure 2 and Table 3), epithelial-mesenchymal transition (EMT) (Figure 3 and Table 4), epithelial adherens junctions signaling (Figure 4), differential expression of cyclins and cell cycle regulation signaling (Figure 5), MYC network (Figure 6), STAT3 pathway (Figure 7 and Table 5), tumor cell microenvironment pathway (Figure 8, Table 6), calcium signaling, human embryonic stem cell pluripotency signaling (Figure 9), and Rho Family GTPases signaling (Figure 10). Also, we conducted data integration of differential gene expression, LEF1 Protein Levels, Wnt Signaling, and Cellular Responses (Figure 11).

### 2.4. Epithelial-Mesenchymal Transition and E-Cadherin/β-Catenin/Wnt Pathway

The AHCY deficient SW480 cell line exhibits significant changes in expression levels of genes that are part of the epithelial-mesenchymal transition (EMT). We observed changes in the lymphoid enhancer-binding factor 1 (LEF1) gene expression levels. Accordingly, we have observed changes in Wnt signaling (Figure 2 and Figure 3, Table 3), which is to be expected as LEF1 is part of the Wnt/β-catenin signaling pathway.

Other changes within the Wnt pathway include WNT6 (Wnt family member 6) with a highly increased expression. Wnt6 is highly conserved in various species, mainly considered to be a member of the β-catenin-dependent Wnt signaling pathway.

Besides the upregulation of LEF1 and its possible regulatory role in Wnt signaling, we have observed the reduced expression of cadherin 12 (CDH12) (Table 3).

Also, we show increased ALCAM expression (activated leukocyte cell adhesion molecule). The expression of ALCAM correlates with the expression of Snail proteins, showing Snail 1 and Snail 2 to be more active (Figure 4, Figure 6 and Figure 11, Table 4 and Table 5). Snail proteins are well-known EMT-Transcription factors, directly binding and suppressing E-cadherin at the proximal CDH1 promoter and remodeling intercellular adhesion.

Also, significant alterations in genes associated with tumor cell invasion were detected, such as significant upregulation of TGFβ1, ROAR, DAB2, BMP6, NOS2, PLXN2, and CADPS exhibited, and significant downregulation of TCF4 (Figure 11).

### 2.5. Differential Expression of Cyclins and Cell Cycle Regulation Signaling

The transcript levels of Cyclin A, Cyclin B, and CDK1 were found to be significantly increased (Figure 5). Cyclin B, in collaboration with CDK1, orchestrates the transition from the G2 phase to the mitotic phase, enabling successful cell division. Accordingly, we observed a higher level of gene expression of RB1, E2F, and TFDP1, which are key regulators of the cell cycle.

### 2.6. MYC, STAT3 and Human Embryonic Stem Cell Pluripotency Signaling

There is an established connection between c-MYC and AHCY. Namely, c-MYC regulates expression of AHCY in human breast cancer cells. Thus, we have identified several proteins that are potentially regulated by MYC (Figure 6). Among them, the expression levels of AHCY, SOX5, SON, OLR1, LRG1, and COL4A1 were found to be significantly associated with MYC activity, specifically in terms of their potential downregulation. MYC exerts its regulatory role by directly binding to the regulatory regions of these genes and modulating their transcriptional activity.

Additionally, upregulation and activation of STAT3, MYC, CDC25A, and BCL2 is shown (Figure 7 and Table 7), suggesting potential implications for enhanced cellular proliferation, survival, and anti-apoptotic responses.

### 2.7. The Tumor Microenvironment Pathway

AHCY-deficient SW480 cells exhibit significant upregulation of genes important for the tumor microenvironment pathway, such as MMP19, MMP24, and CSF2, as well as upregulated activation of PLAU and BCL2 (Figure 8 and Table 6).

We have observed higher expression of TIAM1 and activation of Rac1 (Figure 8). TIAM1 acts as a GEF, a protein that facilitates the exchange of GDP (guanosine diphosphate) for GTP (guanosine triphosphate) on Rac1. 

### 2.8. G Protein-Coupled Receptor (GPCR) Signaling

RNA-seq data revealed an interesting observation regarding the expression of G protein-coupled receptor (GPCR) signaling. We found a downregulation of Adora receptors (GPCR) signaling in the AHCY-deficient cells (Figure 6), suggesting a potential alteration in cellular responses to external stimuli. This downregulation may place the tumor cells under increased stress, making them more competitive for limited resources and driving the activation of mechanisms responsible for cell migration and invasion.

### 2.9. Signaling by Rho Family GTPases

We observed activation of ROCK kinase, which further contributes to cytoskeletal remodeling and cell contractility (Figure 10). Namely, Rho GTPases, including Rho, PIP5K, and ROCK, are key regulators of cytoskeletal dynamics. The Rho signaling cascade also involves downstream effectors, such as FAK (focal adhesion kinase). Additionally, the increased activation of FAK (Figure 9), suggests enhanced focal adhesion turnover and cell-substrate adhesion including cytoskeleton reorganization.

### 2.10. Western Blotting

Western blotting results confirmed efficient lentiviral mediated knockdown of AHCY gene expression in SW480 cells, with significantly decreased AHCY protein levels (Figure 12a).

Also, Western blotting revealed a significant increase in the protein levels of LEF1 in AHCY-deficient SW480 cells when compared to the control cells (Figure 12b). Quantification of band intensities showed an approximately 50% increase in LEF1 protein expression in the AHCY-deficient cells. This observation indicates that the deficiency of AHCY has a direct impact on the expression of LEF1 in SW480 cells. Additionally, we found a significant increase in the expression of STAT3 protein in AHCY-deficient cells compared to control cells (Figure 12c).

## 3. Discussion

In order to imbalance the cellular SAM-to-SAH ratio and inflict changes to the cellular methylation potential, we knocked-out endogenous AHCY in the model cell line SW480, which leads to an accumulation of SAH. The elevated levels of SAH in the AHCY deficient cells may have implications for cellular processes that rely on proper methylation, such as gene expression regulation and epigenetic modifications.

Indeed, we have established a new link between S-adenosylhomocysteine hydrolase and cancer cell signaling by analyzing differentially expressed pathways in AHCY deficient SW480 cells. Namely, after AHCY knockdown, these cells exhibit significantly increased LEF1 protein levels, placing LEF1 into a complex interplay of various signaling pathways and molecular players involved in tumor cell migration and invasion, in particular the Wnt signaling pathway, where the upregulation of LEF1 possibly is disrupting the TCF/LEF transcription factors ratio.

Namely, LEF1 is part of the Wnt/β-catenin signaling pathway, which includes for example genes such as c-Myc, LBH, Oct4, NANOG that have been associated with the upregulation of proteins typically involved in human breast cancer, gastrointestinal tumors, prostate cancer, leukemia, and others [20,21,22,23,24,25,26,27,28,29,30,31,32,33,34,35]. Some of these genes confer stem cell qualities including c-myc, cyclin D1, Oct4, and NANOG, and siRNA-mediated knockdown initiates differentiation, respectively. In addition, c-Myc appears to serve as a master regulator, playing a critical role in in embryonic development and regulating the transcription of genes involved in the cell cycle, and targets molecules involved in the G1/S transition such as CDK2, CDK4, CDC25A, and E2Fs [36].

Cyclin D1 is involved in cell cycle progression, especially in the G1 phase, and is necessary for growth and proliferation. They also serve as downstream effectors of Wnt signaling and are activated by the recruitment of LEF1 to their respective promoter sites. Namely, the promoters for c-myc and cyclin D1 contain LEF1 consensus sequences that allow β-catenin-LEF1 to bind and modulate c-myc and cyclin D1 expression. Other downstream target genes involved in proliferation could be affected as well, such as survin, and VEGF [36,37,38,39]. 

Point mutations of LEF1 located in exons 2 (K86E) and 3 (P106L) of LEF1 result in increased promoter activity and expression for c-myc and cyclin D1, causing increased leukemia cell proliferation [38]. Altered LEF1 expression and function commonly occur in several cancers, such as lung adenocarcinoma, colon cancer, endometrial carcinoma, prostate cancer, and leukemia [38,39]. LEF1 has been reported to promote EMT in cancer cells by activating Wnt/β-catenin signaling, which can in turn activate Notch signaling and GPCR signaling. High LEF1 and low Notch2 expression patterns are associated with tumorigenesis, shorter overall survival time, and higher risk of death in CRC patients. Also, the presence of increased LEF1 is associated with an increased risk for primary colorectal cancer and liver metastasis [40,41,42,43,44,45]. On the other hand, knockdown of LEF1 in colon cancer cells results in various effects on cellular processes such as (a) increased apoptosis compared to control cells in vitro, and reduced tumor growth compared to normal colon cancer cells in vivo, (b) reduced invasiveness via decreased MMP-2 and MMP-9 expression, and (c) changed expression of genes involved in regulation of expression of matrix metalloproteinases such as metallopeptidase 7 (MMP7, a Zn^2+^—dependent proteolytic enzyme) [43]. These studies demonstrate the importance of LEF1 in elucidating typical cancer characteristics, including proliferation, invasion, migration, and viability, amongst a variety of cancer types, and highlight its necessity in propagating these effects. Not only is LEF1 at the center of signaling pathways and mechanisms that initiate and maintain carcinogenesis, the suppression of LEF1 reduces the proliferative and invasive properties of cancer. Similarly, calcium signaling has been shown to regulate Wnt/β-catenin signaling by activating calcium/calmodulin-dependent protein kinase II (CaMKII), which can phosphorylate LEF1 and activate its transcriptional activity. Therefore, LEF1 may indirectly modulate the activity of several of the signaling pathways identified in the RNA-seq analysis by interacting with other key signaling molecules, such as β-catenin, Notch, and GPCRs. This assumption is fostered by finding significantly lower expression levels of TCF4, whereas TCF4 is known to be in a direct interaction with LEF1 known as an interaction TCF/LEF, where overexpressed LEF1 leads to an enhanced tumor cell invasiveness and induces epithelial to mesenchymal transition [43]. Namely, transcription of LEF1 can be directly regulated by TCF4–β-catenin complexes [45,46,47,48]. As LEF-1 is not expressed in the normal colon mucosa [49] but is found in human colorectal cancer [50], a shift of β-catenin binding partners from TCF4 to LEF-1 might occur during carcinogenesis which might enable enhanced epithelial-mesenchymal transition (EMT) and malignant progression.

To conclude: LEF1 is a crucial component of the Wnt signaling pathway. It is a transcription factor that interacts with β-catenin to regulate the expression of Wnt target genes. In the absence of Wnt signaling, β-catenin is typically targeted for degradation. However, when the Wnt pathway is activated, β-catenin accumulates in the nucleus and forms a complex with LEF1. This complex activates the transcription of Wnt target genes, which are involved in various cellular processes, including cell proliferation and differentiation.

The precise mechanisms and consequences of these interactions may vary depending on the cellular context, specific target genes, and extracellular signals present. Further investigations are necessary to fully elucidate the intricate interplay between LEF1 and these canonical and non-canonical pathways in the context of our study, considering their potential impact on gene expression and cellular processes.

### 3.1. Epithelial-Mesenchymal Transition

So far, the hallmark of EMT is the loss of epithelial marker expression, typically indicated by the presence of E-cadherin, with a gain in mesenchymal marker expression such as of N-cadherin and vimentin accompanied by invasive phenotype [51]. Therefore, the E-cadherin/β-catenin/Wnt pathway signaling is pivotal for comprehending the potential consequences of identified gene expression alterations, given the observed changes in Wnt signaling, and its central role in governing cell–cell adhesion and the regulation of cell proliferation—both frequently perturbed within the tumor microenvironment. As we have found, WNT6, highly conserved in various species, with a highly increased expression, and mainly considered to be a member of the β-catenin-dependent Wnt signaling pathway [52], might increase the proliferative ability of colorectal cancer cells (CRC), leading to increased expression of MMP2, which is also involved in the breakdown of the extracellular matrix [53]. In addition, the promoter region of Wnt6 is bound by polymorphic adenoma-like protein 2 (PLAGL2) in the nucleus of CRC cells [54]. PLAGL2, a zinc finger protein derived from the PLAG gene family [54,55,56,57], is a proto-oncogene and a transcription factor. PLAGL2 combines with the Wnt6 promoter and activates the β-catenin-dependent Wnt signaling pathway, thereby stimulating various downstream target genes (such as MMP7, CCND1) and promoting tumor development [54].

E-cadherin together with β-catenin as an adaptor protein establishes links to the actin cytoskeleton. Under physiological conditions, cytoplasmic β-catenin remains in an inactive state by being bound to the APC/GSK3β/Axin/CK1 degradation complex and undergoes phosphorylation for ubiquitination. Wnt signaling inhibits this degradative process by phosphorylating and inhibiting the GSK3β complex. Under conditions that amplify aberrant Wnt signaling, β-catenin translocates into the nucleus and binds to TCF-4/LEF-1 proteins to induce Wnt target genes such as c-Myc, cyclins, MMP, etc., leading to uncontrolled cell proliferation and growth [58]. In the absence of E-cadherin, un-sequestered β-catenin released from the membrane-bound cadherin-catenin complex leads to excess cytoplasmic β-catenin. It has been demonstrated that β-catenin uses the same binding interface to engage TCF and E-cadherin ligands and cadherins have a superior binding affinity. There is a suggestion that as E-cadherin protein is lost, there is excess un-sequestered cytoplasmic β-catenin that escapes degradation and enters the nucleus to bind to TCF and activate the Wnt pathway. In addition to activating downstream Wnt-associated genes, it is also demonstrated that nuclear translocation of β-catenin represses PTEN expression. PTEN is a tumor suppressor and a critical regulator of AKT/MTOR pathway. Thus, the carefully balanced Wnt/β-catenin/E-cad functioning is tipped in favor of uncontrolled cell-proliferation-promoting oncogenesis [47]. In addition, important signaling interactions between E-cadherin and other cellular pathways include RTK/EGFR/MAPK, and the P-120/Rho/RAC pathway, respectively.

Besides the upregulation of LEF1 and being part of the Wnt/β-catenin signaling pathway with possible regulatory roles in aforementioned processes, we have observed the reduced expression of cadherin 12 (CDH12) that has been implicated in promoting increased metastatic potential and cell migration [59]. The loss or downregulation of CDH12 can disrupt the adhesive interactions between cells, leading to a decreased cohesive behavior within the primary tumor and facilitating the detachment of tumor cells from the primary site. This loss of cell adhesion can enhance the migratory capacity of cancer cells, enabling their invasion into surrounding tissues and dissemination to distant sites. Consequently, the decreased expression of CDH12, along with reduced adhesive molecules, may contribute to a more aggressive and metastatic phenotype in cancer cells [59,60,61].

Contributing to potential migratory capacity might be enhanced expression of ALCAM, which is involved in cancer cell migration, in conjunction with the activation of the EMT pathway. ALCAM facilitates interactions between cells and their surrounding environment, and influences cytoskeletal rearrangements, promoting cellular protrusions that expedite directed cell migration. The expression of ALCAM correlates with the expression of SNA1, which is evident in our data [62].

Snail family members (Snail [SNAI1] are well-known EMT-TFs. The EMT-TFs directly bind and suppress E-cadherin at the proximal CDH1 promoter and remodel intercellular adhesion. Snail also suppress other epithelial markers and activate mesenchymal genes, and, as we see in our data, Snail 1 and Snail 2 are more active (Figure 4, Table 5). Furthermore, EMT-TFs are known to reorganize epithelial polarity molecules and impede basement membrane formation to promote pro-invasive circumstances [62,63].

Significant upregulation of expression is found in genes associated with tumor cell invasion such TGFβ1, ROAR, DAB2, BMP6, NOS2, PLXN2, and CADPS, whereas TCF4 was significantly downregulated in AHCY deficient cells. Thus, our data bode well for linking gene activity changes to tumor progression and metastasis, suggesting that, (a) TGFβ1’s elevated activity aligns with invasion promotion, (b) ROAR’s surge suggests heightened invasiveness, (c) DAB2’s rise echoes invasion dynamics, (d) BMP6’s elevation points to tumor progression, (e) NOS2’s increase connects to invasiveness, (f) PLXN2’s upregulation implies migration involvement and, (g) CADPS’s rise aligns with invasiveness [63,64,65,66,67,68,69].

Interestingly, in the context of cell migration and invasion, we found downregulation of Adora receptors (GPCR), suggesting a potential alteration in cellular responses to external stimuli. This downregulation may place the tumor cells under increased stress, making them more competitive for limited resources and driving the activation of mechanisms responsible for cell migration and invasion.

Additional cytoskeletal dynamics are possibly mediated through signaling by Rho Family GTPases, fostered by the observed activation of ROCK kinase, which contributes to cytoskeletal remodeling and cell contractility. Namely, Rho GTPases, including Rho, PIP5K, and ROCK, are key regulators of cytoskeletal dynamics, including downstream effectors, such as FAK (focal adhesion kinase), which exhibits increased activation suggesting an enhanced focal adhesion turnover and cell-substrate adhesion, including cytoskeleton reorganization [70,71]. Thus, the activation of Wnt signaling, STAT3, Rho GTPases, ROCK kinase, FAK, the regulation of the EMT pathway and the upregulation of LEF1 protein collectively might contribute to cytoskeletal reorganization, cell trafficking, and enhanced cell motility, and highlight the complicated network of molecular events involved in the invasive behavior of tumor cells. Calcium signaling can intersect with the non-canonical Wnt pathway, potentially affecting LEF1 through various mechanisms, while LEF1, as a downstream effector of the canonical Wnt pathway, plays a role in regulating epithelial-mesenchymal transition (EMT). Additionally, G protein-coupled receptor (GPCR) signaling can influence the canonical Wnt pathway [72,73,74].

### 3.2. Differential Expression of Cyclins and Cell Cycle Regulation Signaling

Increased transcript levels of Cyclin A, Cyclin B, and CDK1 indicate a potential modulation of cell cycle dynamics in response to AHCY deficiency. Cyclin B, in collaboration with CDK1, orchestrates the transition from the G2 phase to the mitotic phase, enabling successful cell division [75]. The increased expression of Cyclin B and CDK1 implies an augmented drive toward mitosis, possibly reflecting a compensatory mechanism triggered by AHCY deficiency, supported by higher expression of RB1, E2F, and TFDP1, which are key regulators of the cell cycle, being in a more active state in the studied cells. Considering the increased activity of RB1, E2F, and TFDP1, it can be inferred that these molecules are facilitating cell cycle progression.

RB1, when inactive or phosphorylated, releases its inhibitory effect on E2F transcription factors. The active E2F factors, in turn, promote the transcription of genes involved in DNA replication and cell division. TFDP1 interacts with E2F, forming the E2F/TFDP1 complex, which enhances the transcriptional activity of E2F. This complex further promotes the expression of genes required for cell cycle progression [76].

The activation of the E2F/TFDP1 complex leads to the transcription of genes involved in various phases of the cell cycle, such as G1/S transition, S phase, G2 phase, and M phase. However, it is unclear whether the dysregulated cyclin signaling observed in AHCY deficient SW480 cells may be influenced by aberrant LEF1 activity and its interplay with the Wnt pathway.

### 3.3. MYC, STAT3 and Human Embryonic Stem Cell Pluripotency Signaling

Significant perturbations in expression levels have been observed for several proteins that are potentially regulated by MYC, but also MYC itself. Namely, MYC exerts its regulatory role by directly binding to the regulatory regions of distinct genes such as SOX5, SON, OLR1, LRG1, and COL4A1, which have been found to be significantly associated with MYC activity, specifically in terms of their potential downregulation. Proteins SOX5, SON, and OLR1 have been implicated in various aspects of cancer progression and metastasis [77]. SOX5, a paralog of SOX2 within the SOX transcription factor family, may play a significant role in regulating cellular processes in the AHCY deficient SW480 cell line. Furthermore, the presence of such connections in other cellular systems and the known involvement of SOX6 (a paralog of SOX2) in Wnt signaling, strengthens the importance of these findings within the broader context of cell regulation and signaling pathways in the tumor microenvironment. In the case of COL4A1, which is a component of basement membranes, lower expression could potentially disrupt the integrity of basement membranes. Our study also explores the connection between AHCY deficiency and pluripotency signaling in ESCs, focusing on observed upregulation of STAT3, and MYC, together with CDC25A, and BCL2, that suggest potential implications for enhanced cellular proliferation, survival, and anti-apoptotic responses. Also, as MYC, CDC25A, and BCL2 show increased activity, one might argue regarding possible contributions to cell growth, cycle control, and resistance to cell death as shown elsewhere [78,79]. Furthermore, the connection between STAT3 signaling and LEF1 protein levels suggests a potential interplay between these pathways, possibly mediated through crosstalk with the Wnt signaling pathway. Namely, STAT3 activation is crucial for ESC self-renewal by controlling important pluripotency genes like OCT4, NANOG, and SOX2. Additionally, STAT3 influences ESC growth and proliferation by affecting genes involved in the cell cycle, including cyclin D1 and c-myc. The downstream factor MYC also plays a role in ESC development, cell growth, proliferation, and maintenance of pluripotency [80]. Therefore, our analysis also establishes links between AHCY deficiency and pluripotency signaling, shedding new light on the interaction of STAT3 and downstream elements in the HESCPS network.

### 3.4. The Tumor Microenvironment Pathway

Significant changes have been detected in the tumor microenvironment pathway including genes MMP19, MMP24, and CSF2, as well as PLAU, BCL2, TIAM1 and Rac1. These changes might be attributed to the increased LEF1 protein levels, suggesting potential implications for the modulation of the tumor microenvironment in response to AHCY deficiency.

In addition, CSF2, also known as GM-CSF (Granulocyte-Macrophage Colony-Stimulating Factor), is a cytokine that plays a crucial role in the regulation of immune cell development, function, and inflammation. In addition to its immunomodulatory functions, emerging evidence suggests that CSF2 also contributes to cancer cell migration and invasion in the context of cancer metastasis, and indeed we detected highly increased expression of CSF2. In epithelial ovarian cancer cells, the activation of the CSF2/p-STAT3 pathway leads to the enhancement of stem cell-like properties in cancer cells [81].

TIAM1 acts as a GEF, a protein that facilitates the exchange of GDP (guanosine diphosphate) for GTP (guanosine triphosphate) on Rac1. This exchange shifts Rac1 into its active, GTP-bound form, triggering downstream signaling pathways that promote cell migration. Within the existing scientific literature, there is a firmly established comprehension of TIAM1’s integral role in cell proliferation and tumorigenic potential [81,82].

Loss of TIAM1 or RAC1 inhibition induces cell death via BAX/BAK-mediated apoptosis. TIAM1-Nur77 interaction is required for small cell lung cancer (SCLC) cell survival [82,83].

In addition, TIAM1 was elevated in thyroid cancer, and TIAM1 knockdown repressed thyroid cancer cell proliferation and promoted ferroptosis through regulating Nrf2/HO-1 axis. Taken together, these findings may suggest that TIAM1 plays a significant role in the tumor microenvironment signaling pathway by activating cell proliferation and tumorigenic potential [83].

### 3.5. Potential Mechanism of AHCY-KD-Mediated Wnt Activation

The activation of the Wnt signaling pathway in the context of increased LEF1 protein expression and reduced AHCY expression may occur through the following potential mechanism: epigenetic regulation of Wnt pathway genes. Reduced DNA and RNA methylation due to AHCY deficiency can disrupt the epigenetic regulation of genes within the Wnt signaling pathway. Hypomethylation-induced upregulation of Wnt ligands and receptors can result in an increased availability of Wnt ligands, facilitating enhanced binding to cell surface receptors. This series of events initiates intracellular signaling, ultimately culminating in the activation of the Wnt pathway [84]. In summary, the interplay between elevated LEF1 protein levels and reduced AHCY expression likely facilitates the activation of the Wnt signaling pathway through the epigenetic regulation of Wnt pathway genes, such as WNT6, MYC and LEF1 in our example. This intricate mechanism underscores the intricate relationship between transcription factors, methylation potential, and the Wnt pathway in the context of colorectal cancer.

Taken together, our findings provide valuable insights into the complex interplay of various signaling pathways and molecular players involved in tumor cell migration and invasion. Understanding these mechanisms at a molecular level may pave the way for the development of targeted therapeutic interventions aimed at disrupting these pathways and inhibiting tumor metastasis. Further investigations into the precise molecular mechanisms underlying the observed alterations and their implications for tumor progression will be crucial for a comprehensive understanding of tumor biology.

## 4. Materials and Methods

### 4.1. Cell Culture

SW480 cells were obtained from the American Type Culture Collection (ATCC) and cultured in DMEM (Dulbecco’s Modified Eagle’s Medium, Thermo Fisher Scientific, Waltham MA, USA) supplemented with 10% fetal bovine serum (FBS, Thermo Fisher Scientific, Waltham MA, USA) and 1% penicillin-streptomycin (Thermo Fisher Scientific, Waltham, MA, USA) at 37 °C in a humidified atmosphere with 5% CO_2_.

HEK293T cells were obtained from the American Type Culture Collection (ATCC) and cultured according to SW480 cells. 

Cells were sub-cultured every 2–3 days and passages 5–10 were used for all experiments.

### 4.2. Lentivirus Production

HEK293T cells were seeded in a 10-cm dish at a density of 5 × 10^6^ cells/dish and incubated overnight. The cells were transfected with 5 μg of the short hairpin RNA (shRNA) lentiviral vector plasmids shRNA2 and shRNA4 (Merck KGaA, Darmstadt, Germany, both targeting AHCY, respectively, and helper plasmids psPAX2 (3.75 μg) and pMD2.G (1.25 μg) using Lipofectamine 3000 (Thermo Fisher Scientific, Waltham, MA, USA), according to the manufacturer’s instructions. For the production of control cells, we used SHC016 non-target shRNA plasmid (Merck KGaA, Darmstadt, Germany), in combination with psPAX2 and pMD2.G plasmids. After 24 h, the transfection medium was replaced with fresh medium. The supernatant containing functional lentiviral particles was collected 48 h and 72 h post-transfection, pooled, and filtered through a 0.45 μm syringe filter (Merck KGaA, Darmstadt, Germany).

### 4.3. Cell Culture and Antibiotic Resistance Testing

SW480 cells were cultured in DMEM media supplemented with 10% fetal bovine serum (FBS) and 1% penicillin-streptomycin in a humidified incubator at 37 °C and 5% CO_2_. Prior to lentiviral transduction, cells were tested for antibiotic resistance to puromycin using the MTT assay.

Cells were seeded at a density of 2000 cells per well in a 96-well plate and incubated for 24 h to allow for cell attachment. Puromycin was added at various concentrations (0.2, 0.4, 0.6, 0.8, 1 μg/mL) and incubated for 48 h. Following incubation, the media was removed, and the cells were washed with phosphate-buffered saline (PBS). MTT solution (5 mg/mL) was added to each well and incubated for 4 h at 37 °C. The MTT solution was removed, and the formazan crystals were dissolved in dimethyl sulfoxide (DMSO). Absorbance was measured at 570 nm using a microplate reader (Biotek, Agilent Technologies, Santa Clara, CA, USA). The concentration of puromycin that resulted in 50% inhibition of cell growth (IC_50_) was determined using Excel.

### 4.4. Lentiviral Transduction

SW480 cells were seeded in a 6-well plate at a density of 2 × 10^5^ cells/well and incubated overnight. The cells were then transduced with the lentiviral vector containing the gene of interest and an antibiotic resistance gene using polybrene (Merck KGaA, Darmstadt, Germany) at a final concentration of 8 μg/mL. After 24 h, the transduction medium was replaced with a fresh medium containing the appropriate antibiotic (e.g., puromycin) at a concentration of 1 μg/mL. The cells were then cultured for 3 days to allow for the selection of transduced cells.

### 4.5. Transcriptome Profiling—RNA-Seq

Total cell RNA was extracted from 1 × 10^6^ cells using TRIzol^®^ Reagent (Thermo Fisher Scientific, Waltham, MA, USA) following the manufacturer’s instructions. Two different cell passages were used to extract RNA both for shAHCY and shCTRL cells and treated as a biological replicate. RNA quantity was determined using a Qubit 3.0 Fluorometer and Qubit^®^ RNA BR Assay Kit (Thermo Fisher Scientific, Waltham, MA, USA). Agilent 2100 Bioanalyzer and Agilent RNA 6000 Nano Kits (Agilent Technologies, Santa Clara, CA, USA) were used to assess the sample quality. TruSeq Stranded mRNA Library Prep Kit (Illumina Inc., San Diego, CA, USA), NP-202–1001) was used to prepare libraries from 90 ng of total RNA. The collected libraries were analyzed on a Bioanalyzer 2100, diluted to 1.4 p.m., and sequenced on an Illumina NextSeq500 System using NextSeq500/550 High-Output v2 Kit, with 75 cycles (Illumina Inc., San Diego, CA, USA, FC-404–2005). Run setup, direct data streaming, demultiplexing, and analysis were performed at the BaseSpace Sequence Hub (Illumina Inc., San Diego, CA, USA) using the RNA Express BaseSpace App with default analysis parameters. Signaling pathway analysis was done by Ingenuity Pathway Analysis software (IPA, Ingenuity Systems; see http://www.ingenuity.com, accessed from 03/2022 through 03/2023). The IPA Core Analysis was run with the Causal Network analysis option on the uploaded datasets for transcriptome data. Additional relevant parameters include the measurement value type for transcriptome log_2_ (fold change), a cut-of range: −0.5–0.5; focus on: both up/down-regulated, and species: human. The *p*-value was calculated using the right-tailed Fisher’s exact test.

### 4.6. Western Blotting

Whole-cell proteins were obtained by cell scraping in cold lysis buffer. The pellet is resuspended in 300 µL of a previously prepared cell lysis buffer RIPA (150 mM NaCl, 50 mM Tris, 0.1% SDS, 0.5% sodium deoxycholate (DOC), 1% NP-40) supplemented with protease inhibitors cOmplete™ Mini Protease Inhibitor Cocktail (Merck KGaA, Darmstadt, Germany) and phosphatase inhibitor sodium orthovanadate Na_3_VO_4_ (Thermo Scientific, Waltham, MA, USA) at a final concentration of 1 mM. Following sonication on ice (XL2000 Microson, 5.5 settings, Misonix, Farmingdale, NY, USA). After centrifugation at 14,000 rpm at +4 °C. The protein concentration in the supernatant was determined using a Pierce BCA Protein Assay Kit (Thermo Fisher Scientific, Waltham, MA, USA). Proteins were separated by SDS-PAGE electrophoresis and transferred onto nitrocellulose membranes using the Trans-Blot^®^ TurboTM Transfer System (Bio-Rad Laboratories Inc., Hercules, CA, USA) and Mini Nitrocellulose Transfer Packs (Bio-Rad Laboratories Inc., Hercules, CA, USA), according to the manufacturer’s recommendations with turbo settings for transferring proteins of a wide range of molecular weights. To verify the successful transfer and quantify total proteins on the membrane, the proteins were briefly stained with a Ponceau S solution (0.1% Ponceau S, 5% acetic acid) and then rinsed with TBS-T buffer (Tris Buffered Saline Tween; 50 mM Tris-HCl (pH 7.5), 150 mM NaCl, 0.1% Tween-20). The membrane was then blocked with blocking buffer (5% non-fat milk powder in TBS-T, Merck KGaA, Darmstadt, Germany) for 1 h at room temperature (RT) and washed 3 times for 15 min with TBS-T buffer (Merck KGaA, Darmstadt, Germany). The appropriate primary antibodies, e.g., anti-AHCY (ab134966 Abcam, Cambridge, UK), anti-LEF1 (sc-374522, Santa Cruz Biotechnology, Dallas, TX, USA), anti-STAT3 (sc-8019, Santa Cruz Biotechnology, Dallas, TX, USA), and anti-GAPDH (ab9458, Abcam, Cambridge, UK), were diluted in blocking buffer according to the manufacturer’s recommendation, and the membranes were incubated with it for 2 h at RT or overnight at +4 °C, respectively. The membranes were washed 3 times for 15 min with wash buffer, and the mouse IgG Fc binding protein Horseradish Peroxidase (HRP) conjugated secondary antibody (sc-525409, Santa Cruz Biotechnology, Dallas, TX, USA), diluted in blocking buffer according to the manufacturer’s recommendation, was added for incubation at RT for 1 h for STAT3 and LEF1 membranes. Accordingly, the HRP conjugated secondary Goat Anti-Rabbit IgG H&L antibody (ab6721, Abcam, Cambridge, UK) was used for AHCY and GAPDH detection, respectively. Afterwards, the membranes were washed 3 times for 15 min with wash buffer.

The chemiluminescent signal was developed using the ClarityTM Western ECL Blotting Substrate (Bio-Rad Laboratories Inc., Hercules, CA, USA) kit according to the manufacturer’s recommendation and detected using Alliance Q9 Mini (UVITEC, Cambridge, UK). Densitometry analysis of the signal on the membrane images was performed using the ImageJ software (https://imagej.net/ij/).

### 4.7. Determination of SAM and SAH by LC–MS/MS

Liquid chromatography linked to tandem mass spectrometry (LC-MS/MS) was deployed for the determination of S-adenosylmethionine (SAM) and S-adenosylhomocysteine (SAH) in human cells as a modification of a previously published method by the Kozich laboratory [85]. Namely, instead of using perchloric acid, we modified procedures in favor of ammonium formate as described extensively in Belužić et al. (2018) [86]. 

### 4.8. IPA Core Analysis

The preprocessed gene expression data were uploaded into IPA for pathway analysis using the Core Analysis module. Core Analysis integrates known biological pathways, molecular networks, and functional annotations to analyze the input data.

The Core analysis was run with the Causal Network analysis option on the uploaded datasets for transcriptome data, providing single datasets. 

Molecule identification: IPA mapped the gene symbols from the uploaded data onto its knowledge base to identify the corresponding molecules. This step aimed to ensure that the input genes were correctly annotated and matched with the existing biological information.

Pathway Analysis: IPA performed pathway enrichment analysis using Fisher’s exact test to determine the statistical significance of pathway enrichment based on the input gene expression patterns. A *p*-value threshold of (−0.5, 0.5) was used to identify significantly enriched pathways. Functional analysis: IPA conducted functional analysis to identify the biological functions, diseases, and upstream regulators associated with the input dataset. This analysis involved the prediction of activation or inhibition of regulatory molecules based on the input data and known downstream effects.

Interpretation and Visualization: IPA’s visualization tools, including pathway maps, network diagrams, and functional analysis results, were used to interpret and visualize the results of the pathway analysis. These visualizations aided in understanding the underlying biology and generating hypotheses.

Statistical Analysis in IPA: Statistical significance in pathway enrichment and functional analysis results was determined using appropriate statistical tests: Fisher’s exact test and z-score calculation. Multiple testing corrections Bonferroni were applied to control for false discovery rate where applicable.

### 4.9. Statistical Analysis 

All data are presented as mean ± standard deviation (SD) of at least three independent experiments. Statistical analysis was performed using GraphPad Prism software (version 9.0.1) and differences between groups were analyzed using one-way ANOVA followed by Tukey’s multiple comparison test. A *p*-value of less than 0.05 was considered statistically significant.

## 5. Conclusions

Our study investigated the impact of the knockdown of S-adenosyl homocysteine hydrolase (AHCY) on gene expression and subsequent changes on signaling pathways in the model cell line SW480.

Thus, we show that AHCY deficiency affects the levels of LEF1 protein in SW480 cells, leading to metabolic and signaling shifts causing gene expression changes with broad effects on Wnt signaling, EMT, cell cycle regulation signaling, and the tumor microenvironment pathway with potential implications for enhanced proliferation, cell invasion and metastasis:Methylation potential: Significant alteration in the SAM/SAH ratio is observed between shAHCY (AHCY-deficient) and shCTRL (control) cells, favoring SAH. Since SAH acts as an inhibitor of methylation enzymes, alterations in the SAM/SAH ratio can disrupt the normal methylation regulation within cells. This, in turn, can affect the regulation of gene expression, as methylation plays a crucial role in epigenetic gene control. AHCY deficiency’s impact on methylation capacity may have downstream effects on gene expression, possibly influencing the Wnt signaling pathway and the role of LEF1 and the formation of the β-catenin/LEF1 complex, which is crucial for Wnt target gene regulation in SW480 tumor cells. Further research is needed to explore the intricate interplay between AHCY deficiency, Wnt signaling, and LEF1 in the context of CRC.Calcium Signaling: Calcium signaling can intersect with the non-canonical Wnt pathway through various mechanisms. Calcium ions can modulate the activity of Wnt signaling components, including LEF1, by affecting β-catenin stability, the interaction between β-catenin and LEF1, or downstream signaling events. The PCP pathway after Wnt activation is also responsible for gene expression regulation but also for cell cytoskeleton remodeling, together with ROCK and JNK kinases. The WNT/Ca^2+^ pathway is associated with muscle contraction, gene transcription, and enzyme activation and activates both β-catenin-dependent and β-catenin-independent pathways.Epithelial-Mesenchymal Transition (EMT): LEF1, as a downstream effector of the canonical Wnt pathway, can participate in the regulation of EMT. EMT is a dynamic process involved in tissue remodeling and cancer progression. Activation of the canonical Wnt pathway, including the involvement of LEF1, has been linked to the induction or maintenance of EMT programs.G Protein-Coupled Receptor (GPCR) Signaling: GPCR signaling can intersect with the canonical Wnt pathway through various mechanisms. Wnt ligands can be activated by GPCRs kinases, leading to the activation of downstream signaling cascades, which can modulate the canonical Wnt pathway and potentially influence the activity of LEF1, and we see in our RNAseq data lower differential expression of GPCRs.

In summary, the canonical Wnt pathway and LEF1 indicate interplay on several signaling levels and demonstrate that LEF1 plays a crucial role in cancer survival and activity. Further investigation of LEF1 in the context of cancer progression may contribute to the identification of new therapeutic targets for smart medicine development for CRC patients.

## Figures and Tables

**Figure 1 ijms-24-16102-f001:**
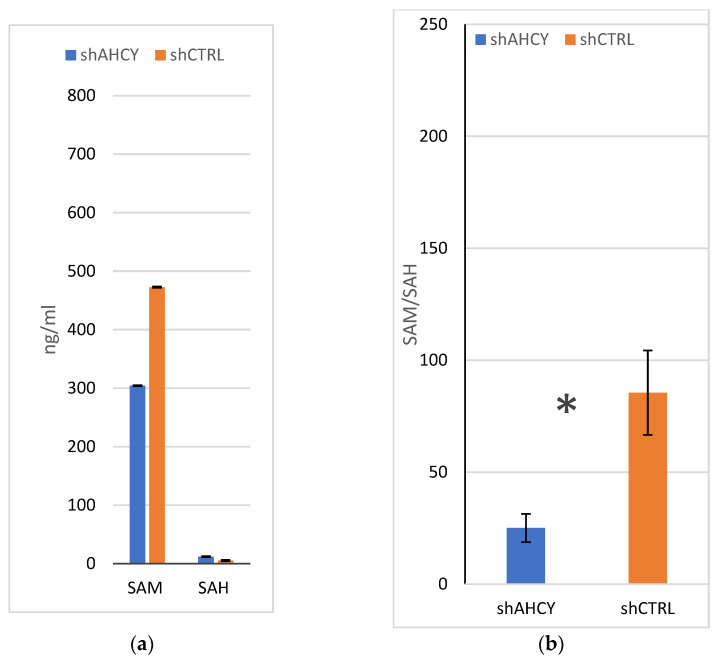
SAM/SAH measurements (**a**) Levels of SAM and SAH (ng/mL) and their ratio (**b**) SAM/SAH in the lysates of AHCY-silenced and control cells, as measured by LC-MS/MS. ±SD is represented as vertical line and is based on three independent measurements. Statistical significance was analyzed using Student’s *t* tests at a * *p* < 0.05.

**Figure 2 ijms-24-16102-f002:**
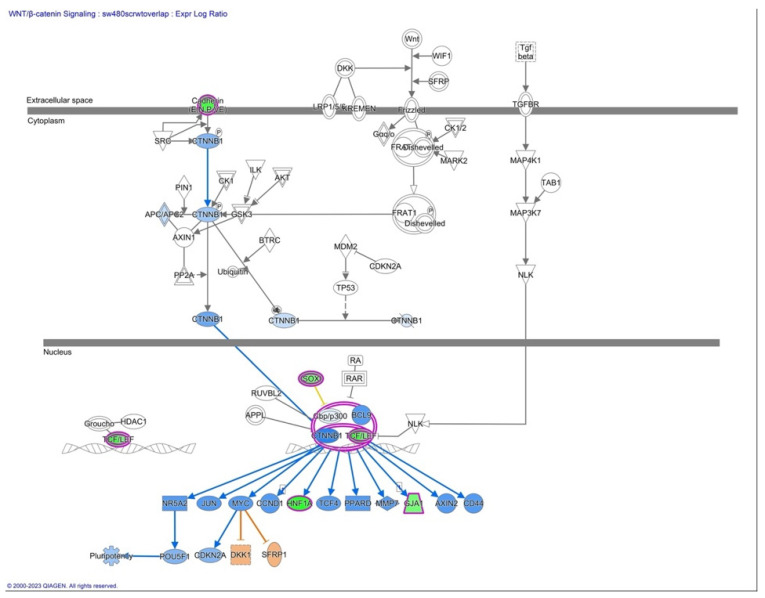
Wnt Signaling pathway. The diagram provides an overview of the major components and interactions within the Wnt signaling pathway such as LEF1 and TCF/LEF1 complex. The color assignments in IPA are based on statistical analyses, such as the z-score. The z-score compares the observed gene expression changes in a given dataset to a reference dataset, assessing the significance and direction of those changes. Positive z-scores indicate upregulation = red, orange = activated, whereas negative z-scores indicate downregulation = green, blue = inhibited, and z-scores close to zero indicate no significant change = no color, and purple indicates interactions of two or more factors. The color scheme employed in this figure is consistent with the color scheme utilized in all presented figures.

**Figure 3 ijms-24-16102-f003:**
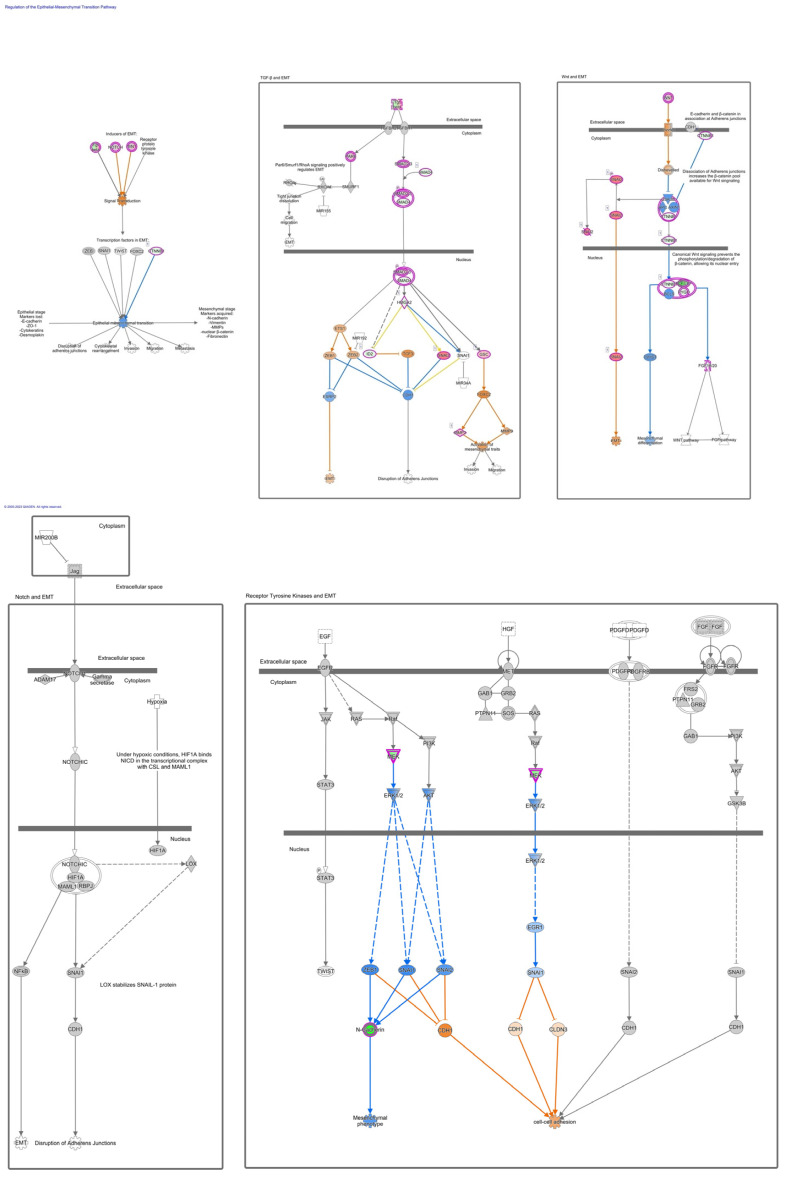
IPA core analysis of the regulation of the Epithelial-Mesenchymal Transition Pathway in SW480 AHCY deficient cells, revealing the involvement of LEF1, a functional transcription factor forming part of the TCF/LEF complex, thereby exerting regulatory control over the expression of genes crucial for EMT.

**Figure 4 ijms-24-16102-f004:**
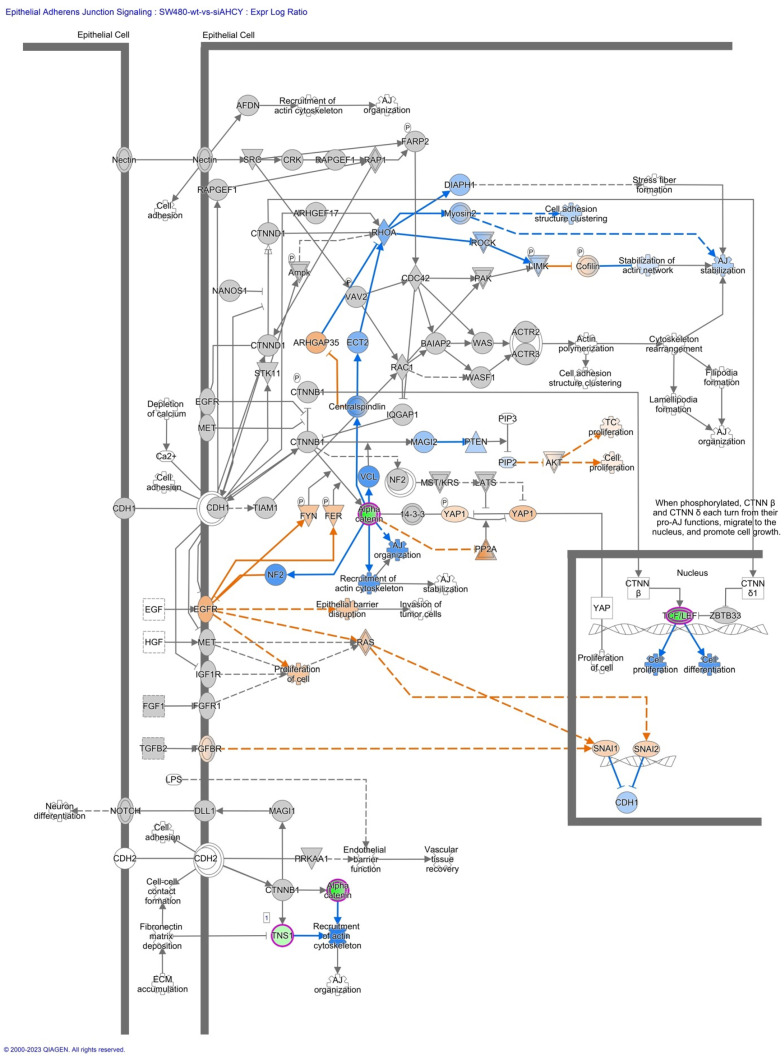
IPA Core analysis of the Epithelial Adherens Junctions Signaling in SW480 AHCY deficient cells with increased expression of LEF1 protein. The accompanying Table 5 provides valuable insights into the transcriptional alterations observed in pivotal genes associated with epithelial adherens junctions signaling following differential expression analysis such as: CDH1, CDH2, TCF/LEF1. Notably, there are changes in gene expression observed in the TCF/LEF complex, mirroring the patterns seen in the Wnt signaling pathway and the regulation of the epithelial-mesenchymal transition pathway.

**Figure 5 ijms-24-16102-f005:**
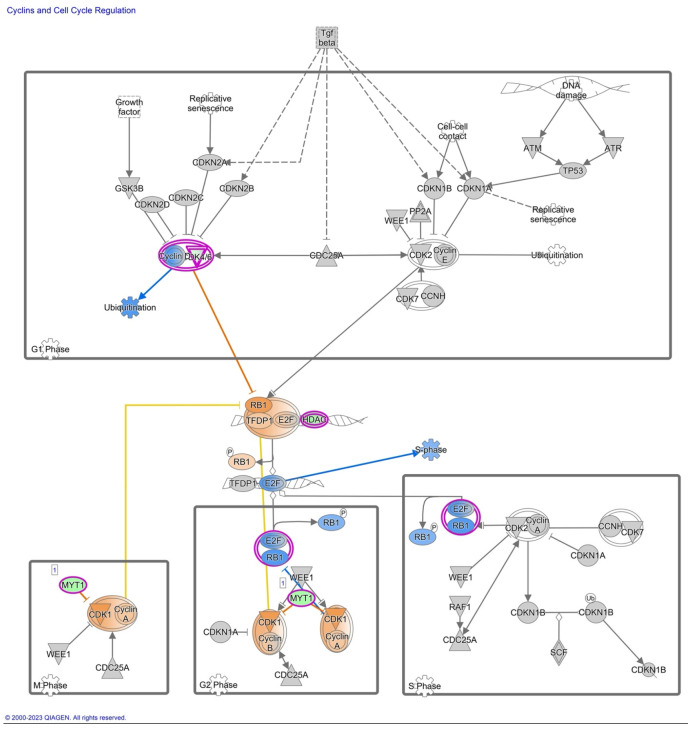
IPA Core analysis of Differential Expression of Cyclins and Cell Cycle Regulation Signaling in SW480 AHCY deficient cells with increased expression of LEF1 protein. Notably, the transcript levels of Cyclin A, Cyclin B, and CDK1 were found to be significantly activated, indicating a potential modulation of cell cycle dynamics in response to AHCY deficiency. Cyclin B, in collaboration with CDK1, orchestrates the transition from the G2 phase to the mitotic phase, enabling successful cell division. The heightened expression of Cyclin B and CDK1 implies an augmented drive toward mitosis, possibly reflecting a compensatory mechanism triggered by AHCY deficiency.

**Figure 6 ijms-24-16102-f006:**
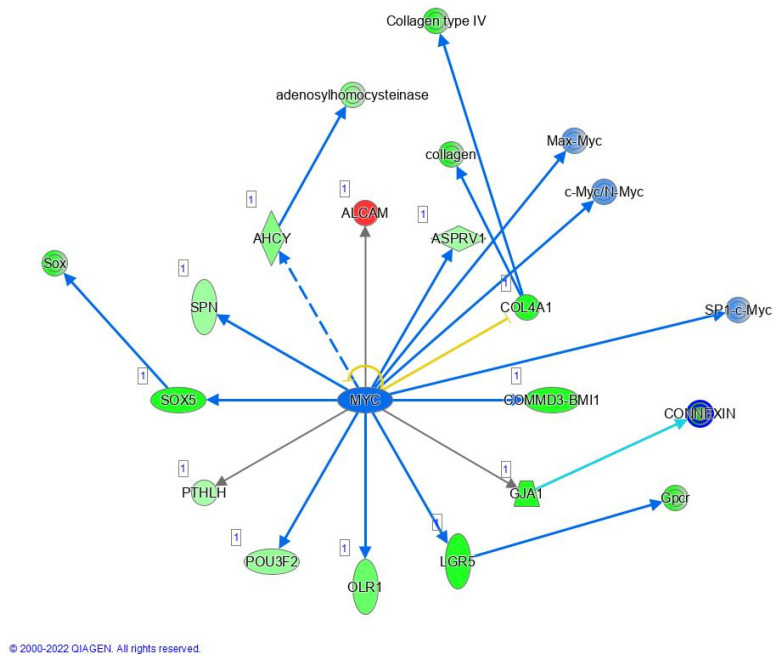
MYC network; IPA Analysis indicates MYC dependent regulation of several proteins including, AHCY, SOX5, SON, OLR1, LGR5, and COL4A1, which expression levels were found to be significantly associated with MYC activity, indicating MYC as a potential master regulator.

**Figure 7 ijms-24-16102-f007:**
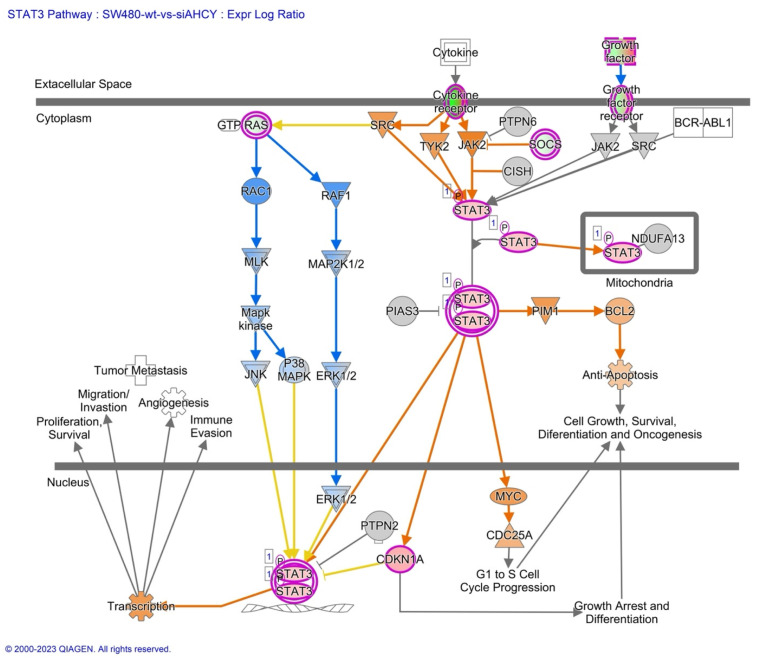
STAT3 Signaling in SW480 AHCY deficient cells. The figure summarizes the upregulation and activation of STAT3, MYC, CDC25A, and BCL2, highlighting potential implications for cellular responses and signaling crosstalk.

**Figure 8 ijms-24-16102-f008:**
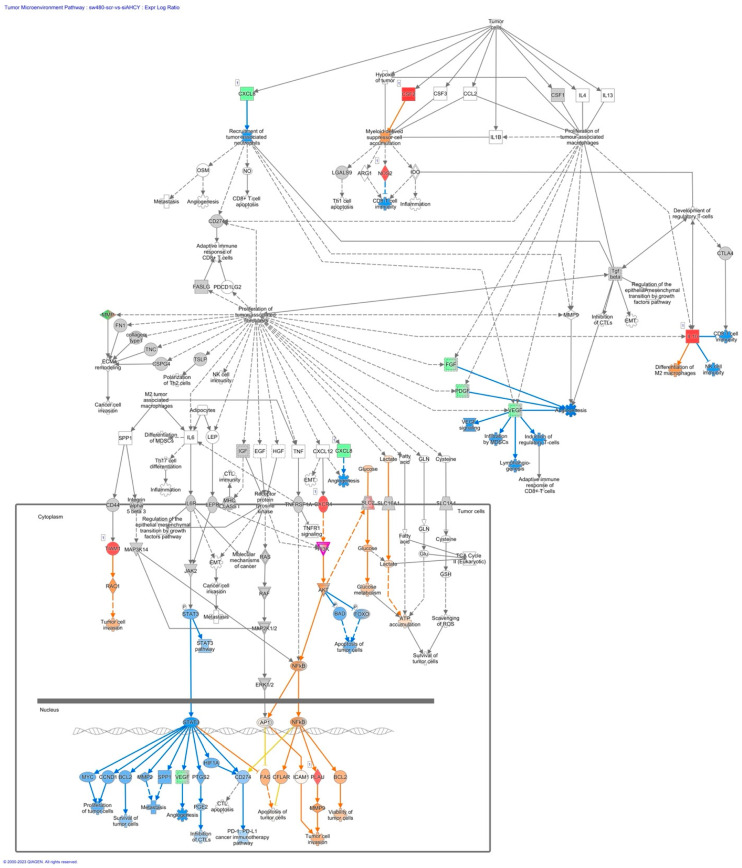
IPA Core analysis of the Tumor Microenvironment Pathway in SW480 AHCY deficient cells with increased expression of LEF1 protein.

**Figure 9 ijms-24-16102-f009:**
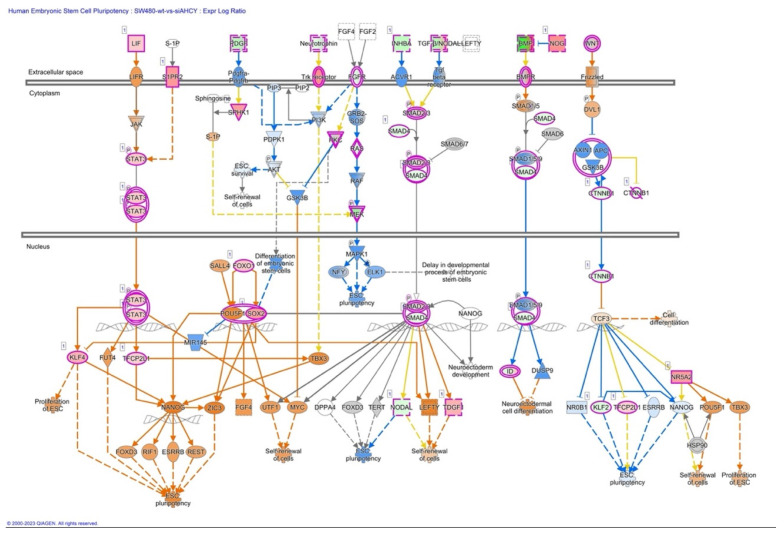
Human Embryonic Stem Cell Pluripotency Signaling. STAT3 plays a critical role in regulating the self-renewal and proliferation of embryonic stem cells (ESCs). The figure illustrates the involvement of STAT3 in maintaining the pluripotent state of ESCs and promoting their proliferation, highlighting key downstream effectors and signaling pathways.

**Figure 10 ijms-24-16102-f010:**
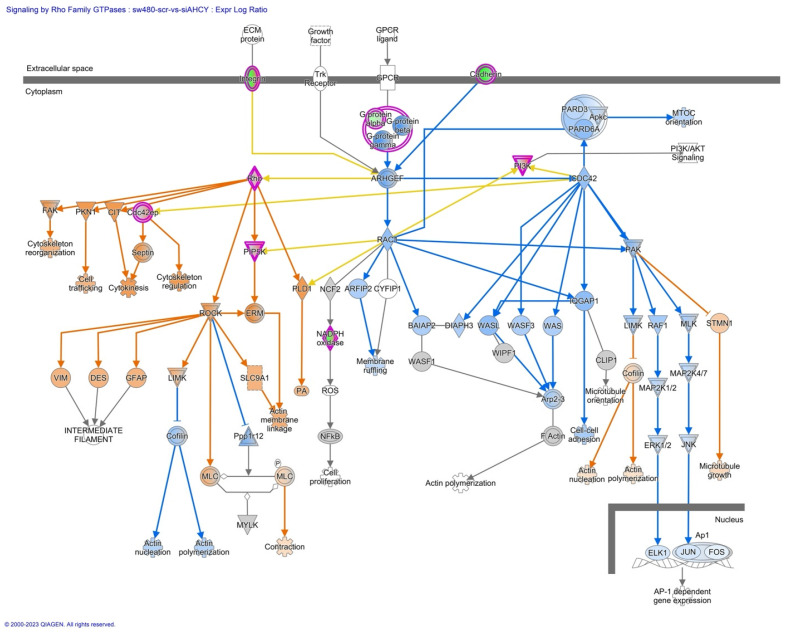
IPA Core analysis of signaling by Rho Family GTPases in SW480 AHCY deficient cells with increased expression of LEF1 protein. The figure depicts the increased activation of cytoskeletal reorganization, cell trafficking, and migration/invasion-related processes in response to AHCY deficiency and increased levels of LEF1 protein. Rho GTPases, including Rho, PIP5K, and ROCK, are shown as key regulators of cytoskeletal dynamics.

**Figure 11 ijms-24-16102-f011:**
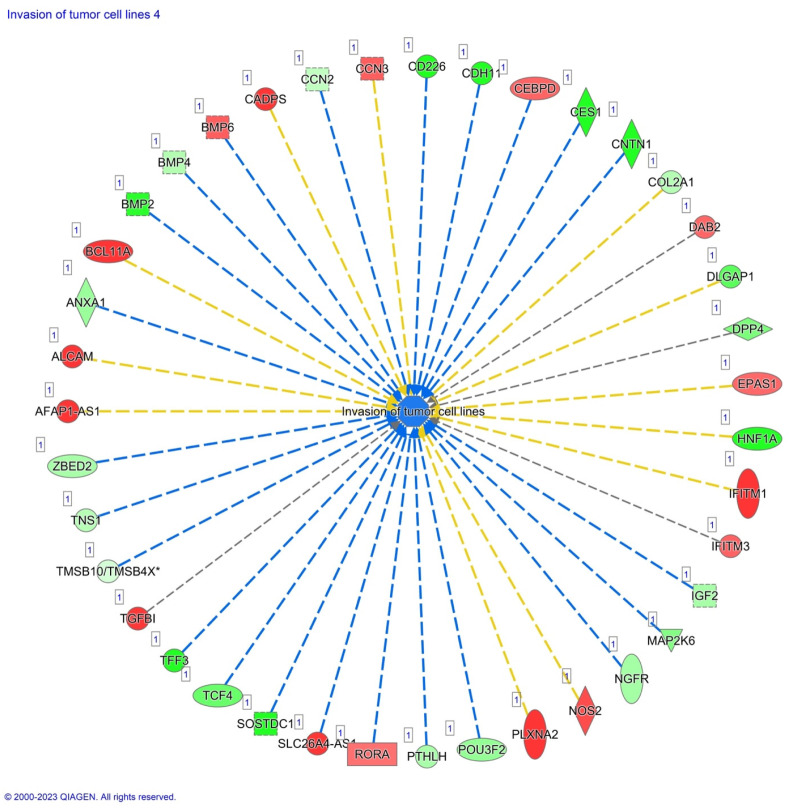
Integration of Differential Gene Expression, LEF1 Protein Levels, Wnt Signaling, and Cellular Responses in AHCY-deficient SW480 Cells. Differential gene expression analysis in AHCY-deficient SW480 cells revealed significant alterations in genes associated with tumor cell invasion. TGFβ1, ROAR, DAB2, BMP6, NOS2, PLXN2, and CADPS exhibited significant upregulation, while TCF4 and AHCY were significantly downregulated. These gene expression changes were connected with increased LEF1 protein levels, activated Wnt signaling, and potential implications for enhanced cell invasion and proliferation through the upregulation of Cyclin A and Cyclin B. Top of FormBottom of Form.

**Figure 12 ijms-24-16102-f012:**
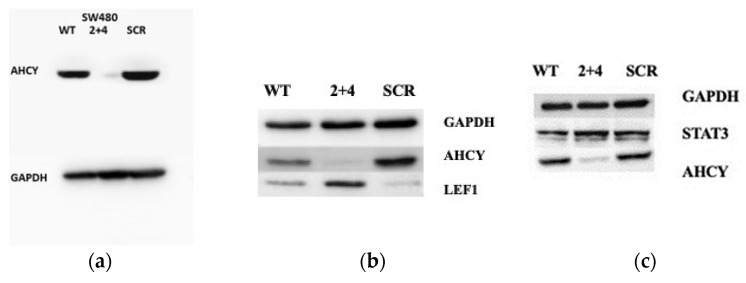
Western blotting results. For all experiments, GAPDH was used as a loading control, and detected by a rabbit polyclonal antibody (ab9458, Abcam). Then, 30 μg of whole cell proteins retrieved from SW480 AHCY-deficient or SW480 control cells were loaded per well. (**a**) Detection of AHCY protein (**b**) Detection of LEF1 protein, (**c**) Detection of STAT3 protein 2 + 4 signifies the group of cells in which AHCY expression has been silenced, while SCR represents the control group with normal AHCY expression. Western blot analysis of LEF1 protein expression confirmed the transcriptomic data predictions and revealed increased LEF1 protein in AHCY-deficient cells.

**Table 1 ijms-24-16102-t001:** Summary of IPA analysis; Molecular and Cellular Functions wt-vs-siAHCY.

Name	*p*-Value Range	Molecules
Cellular movement	6.00 × 10^−4^–2.48 × 10^−13^	73
Cell death and survival	6.18 × 10^−4^–2.31 × 10^−8^	66
Cellular development	5.77 × 10^−4^–2.45 × 10^−7^	76
Cellular growth and proliferation	5.77 × 10^−4^–2.45 × 10^−7^	70
Cell morphology	4.14 × 10^−4^–3.07 × 10^−7^	48

**Table 2 ijms-24-16102-t002:** Summary of IPA analysis; Molecular and Cellular Functions scr-vs-siAHCY.

Name	*p*-Value Range	Molecules
Cellular movement	2.11 × 10^−12^–5.74 × 10^−41^	523
Cell death and survival	3.76 × 10^−13^–8.59 × 10^−28^	553
Cellular functions and maintenance	2.30 × 10^−12^–1.96 × 10^−25^	520
Cellular development	1.51 × 10^−8^–3.38 × 10^−23^	605
Cellular growth and proliferation	1.11 × 10^−4^–8.38 × 10^−23^	600

**Table 3 ijms-24-16102-t003:** Differentially Expressed Genes in Wnt Signaling Pathway based on acquired RNAseq data. The table is summarizing the differentially expressed genes identified in the Wnt signaling pathway using IPA (ingenuity pathway analysis) software after performing differential expression analysis. The table provides insights into the transcriptional changes observed in key genes associated with Wnt signaling in SW480 AHCY deficient cells.

Symbol	Expr Log Ratio	q-Value	Type(s)
CDH12	−11.706	6.97 × 10^−13^	Other
HNF1A	−13.408	3.71 × 10^−17^	Transcription regulator
MAP2K6	−6.754	2.01 × 10^−39^	kinase
TCF4	−3.915	2.45 × 10^−3^	Transcription regulator

**Table 4 ijms-24-16102-t004:** Differentially Expressed Genes in Regulation of the Epithelial-Mesenchymal Transition Pathway. The table provides insights into the transcriptional changes observed in key genes associated with Regulation of the epithelial-mesenchymal transition pathway under AHCY deficient conditions. Differentially expressed changes concerning the TCF/LEF complex, similar as in the Wnt signaling pathway analysis are present.

Symbol	Expr Log Ratio	q-Value	Type(s)
APC2	3.428	1.35 × 10^−2^	Enzyme
CDH12	−11.706	6.97 × 10^−13^	other
DKK1	−3.416	2.55 × 10^−5^	Growth factor
DKK3	3.316	5.54 × 10^−8^	Cytokine
DKK4	−3.791	1.81 × 10^−8^	Other
FZD7	4.134	5.66 × 10^−20^	G-protein coupled receptor
GJA1	−5.544	5.66 × 10^−6^	Transporter
HNF1A	−13.408	3.71 × 10^−17^	Transcription regulator
POU5F1	−5.964	9.76 × 10^−3^	Transcription regulator
RARB	−7.753	1.00 × 10^−3^	Nuclear receptor
SFRP5	3.362	5.03 × 10^−12^	Transmembrane receptor
SOX5	−8.898	3.15 × 10^−6^	Transcription regulator
SOX6	−3.842	3.33 × 10^−6^	Transcription regulator
TCF4	−3.915	2.45 × 10^−3^	Transcription regulator
TLE1	4.455	4.53 × 10^−11^	Transcription regulator
TLE4	4.145	2.28 × 10^−25^	Transcription regulator
WNT6	4.078	1.28 × 10^−21^	other

**Table 5 ijms-24-16102-t005:** The table presents a systematic analysis of the diverse functions of the Epithelial Adherens Junctions Signaling based on RNAseq data and IPA Core analysis. It highlights roles of epithelial adherens junctions signaling in cellular processes such as: cell adhesion, cell to cell contact formation, and remodeling of actin cytoskeleton.

From Molecule(s)	Relationship Type	To Molecules
14-3-3	protein–protein interactions	YAP1
AFDN	causation	Recruitment of actin cytoskeleton
AKT	causation	Cell proliferation
AKT	causation	TC proliferation
ARHGAP35	inhibition	RHOA
ARHGEF17	activation	RHOA
α-catenin	activation	Central spindling
α-catenin	activation	NF2
α-catenin	activation	VCL
α-catenin	causation	AJ organization
α-catenin	causation	Recruitment of actin cytoskeleton
α-catenin	inhibition	PP2A
α-catenin	protein–protein interactions	14-3-3
α-catenin	protein–protein interactions	NF2
α-catenin	protein–protein interactions	VCL
Ampk	activation	RHOA
Arp2-3	causation	Actin polymerization
Arp2-3	membership	ACTR2
Arp2-3	membership	ACTR3
BAIAP2	activation	WAS
BAIAP2	activation	WASF1
BAIAP2	protein–protein interactions	WASF1
CDC42	activation	BAIAP2
CDC42	activation	PAK
CDC42	activation	WAS
CDC42	inhibition	IQGAP1
CDC42	protein–protein interactions	IQGAP1
CDH1	activation	RAPGEF1
CDH1	activation	STK11
CDH1	causation	Cell adhesion
CDH1	inhibition	EGFR
CDH1	inhibition	IGF1R
CDH1	inhibition	MET
CDH1	protein–protein interactions	RAPGEF1
CDH2	activation	PRKAA1
CDH2	protein–protein interactions	CDH2
CDH2	protein–protein interactions	CTNNB1
CDH2	protein–protein interactions	PRKAA1
CRK	activation	RAPGEF1
CTNNB1	activation	α-catenin
CTNNB1	activation	CDH1
CTNNB1	activation	MAGI1
CTNNB1	activation	MAGI2
CTNNB1	activation	NF2
CTNNB1	activation	TNS1
CTNNB1	molecular cleavage	CDH1
CTNNB1	protein–protein interactions	α-catenin
CTNNB1	protein–protein interactions	CDH1
CTNNB1	protein–protein interactions	MAGI1
CTNNB1	protein–protein interactions	MAGI2
CTNNB1	protein–protein interactions	TNS1
CTNNB1	reaction	α-catenin, FER
CTNNB1	reaction	α-catenin, FYN
CTNNB1	reaction	CTNNB1, EGFR; MET
CTNNB1	reaction	MAGI2, VCL
CTNND1	activation	CDH1
CTNND1	molecular cleavage	CDH1
CTNND1	protein–protein interactions	CDH1
CTNND1	protein–protein interactions	RHOA
CTNND1	reaction	CTNND1, CDH1
CTNND1	reaction	CTNND1, NANOS1
CTNND1	translocation	CTNND1
CTNN,β-CDHE/N	activation	ARHGEF17
CTNN,β-CDHE/N	activation	TIAM1
CTNN,β-CDHE/N	causation	Cell adhesion
CTNN,β-CDHE/N	causation	Cell-cell contact formation
CTNN,β-CDHE/N	membership	CDH1
CTNN,β-CDHE/N	membership	CDH2
CTNN,β-CDHE/N	membership	CTNNB1
CTNN,β-CDHE/N	membership	CTNND1
CTNN,β-CDHE/N	protein–protein interactions	CDH1
Ca^2+^	activation	CDH1
Ca^2+^	chemical–protein interactions	CDH1
Central spindlin	activation	ECT2
Central spindlin	inhibition	ARHGAP35
Cofilin	causation	Stabilization of actin network
DIAPH1	causation	Stress fiber formation
DLL1	activation	NOTCH
ECT2	activation	RHOA
EGF	activation	EGFR
EGFR	activation	FER
EGFR	activation	FYN
EGFR	activation	RAS
EGFR	causation	Epithelial barrier disruption
EGFR	causation	Proliferation of cell
EGFR	inhibition	CTNND1
EGFR	phosphorylation	CTNND1
EGFR	phosphorylation	FER
EGFR	phosphorylation	FYN
FARP2	activation	CDC42
FGF1	activation	FGFR1
FGFR1	activation	RAS
HGF	activation	MET
IGF1R	causation	Proliferation of cell
IQGAP1	inhibition	CTNNB1
IQGAP1	protein–protein interactions	CTNNB1
LATS	inhibition	YAP1
LIMK	inhibition	Cofilin
LIMK	phosphorylation	Cofilin
LPS	causation	Endothelial barrier function
MAGI1	activation	DLL1
MAGI1	protein–protein interactions	DLL1
MAGI2	activation	PTEN
MAGI2	molecular cleavage	PTEN
MAGI2	protein–protein interactions	PTEN
MER-WWC1-FRMD6	activation	MST/KRS
MER-WWC1-FRMD6 membership	membership	NF2
MET	activation	RAS
MET	causation	Proliferation of cell
MET	inhibition	CDH1
MET	phosphorylation	CDH1
MST/KRS	activation	LATS
Myosin2	causation	AJ stabilization
Myosin	causation	Cell adhesion structure clustering
NF2	inhibition	EGFR
NOTCH	causation	Neuron differentiation
Nectin	activation	AFDN
Nectin	activation	SRC
Nectin	causation	Cell adhesion
Nectin	protein–protein interactions	AFDN
Nectin	protein–protein interactions	Nectin
Nectin	protein–protein interactions	SRC
PAK	activation	LIMK
PAK	phosphorylation	LIMK
PIP2	inhibition	AKT
PIP3	reaction	PIP2 PTEN
PRKAA1	causation	Endothelia barrier function
RAC1	activation	BAIAP2
RAC1	activation	PAK
RAC1	activation	WASF1
RAC1	inhibition	IQGAP1
RAC1	protein–protein interactions	IQGAP1
RAP1	activation	CTNND1
RAP1	activation	FARP2
RAPGEF1	activation	RAP1
RAS	expression	SNAI1
RAS	expression	SNAI2
RHOA	activation	DIAPH1
RHOA	activation	Myosin2
RHOA	activation	ROCK
ROCK	activation	LIMK
ROCK	phosphorylation	LIMK
SNAI1	expression	CDH1
SNAI2	expression	CDH1
SRC	activation	CRK
SRC	activation	FARP2
SRC	activation	VAV2
SRC	phosphorylation	FARP2
SRC	phosphorylation	VAV2
STK11	activation	Ampk
STK11	phosphorylation	Ampk
TCF/LEF	causation	Cell differentiation
TCF/LEF	causation	Cell proliferation
TGFB2	activation	TGFBR
TGFBR	expression	SNAI1
TIAM1	activation	RAC1
TNS1	causation	Recruitment of actin cytoskeleton
VAV2	activation	CDC42
VAV2	activation	RAC1
WAS	activation	Arp2-3
WASF1	activation	Arp2-3
YAP1	reaction	YAP1 LATS
YAP1	reaction	YAP1 PP2A
ZBTB33	inhibition	TCF/LEF

**Table 6 ijms-24-16102-t006:** Differentially Expressed Genes in the Tumor Cell Microenvironment Pathway in AHCY-Downregulated SW480 Cells. The table summarizes the differentially expressed genes, highlighting upregulated and downregulated genes involved in extracellular matrix re-modeling, immune cell recruitment, cell migration, and cell survival. Significant changes in genes associated with the tumor cell microenvironment pathway are revealed.

Symbol	Expr Log Ratio	q-Value	Type(s)
CSF2	−4.435	7.37 × 10^−13^	Cytokine
CXCLR8	−3.762	6.67 × 10^−10^	Cytokine
CXCR4	3.8	2.03 × 10^−25^	G-protein coupled receptor
FGF21	−3.51	4.09 × 10^−3^	Growth factor
IL10	4.254	4.40 × 10^−14^	Cytokine
MMP16	−6.32	4.43 × 10^−3^	Peptidase
MMP17	−3.808	9.57 × 10^−9^	Peptidase
MMP19	3.196	1.33 × 10^−3^	Peptidase
MMP24	3.114	9.03 × 10^−7^	Peptidase
NOS2	3.881	1.34 × 10^−19^	Enzyme
PDGFC	−3.482	1.34 × 10^−6^	Growth factor
PIK3R5	4.778	1.5 × 10^−2^	Kinase
PLAU	3.487	6.53 × 10^−11^	Peptidase
SLC2A3	3.254	1.67 × 10^−12^	Transporter
TIAM1	3.836	5.84 × 10^−27^	other

**Table 7 ijms-24-16102-t007:** The table presents a systematic analysis of the diverse functions of STAT3 signaling based on RNAseq data and IPA Core analysis. It highlights roles of STAT3 signaling in cellular processes such as proliferation, survival, and differentiation.

From Molecule(s)	Relationship Type	To Molecules(s)
BCL2	Causation	Anti-Apoptosis
BCR-ABL1	activation	STAT3
CDKN1A	Inhibition	Stat3-Stat3
Cytokine	activation	Cytokinereceptor
Cytokine	protein–protein interactions	Cytokinereceptor
Cytokinereceptor	activation	JAK2
Cytokinereceptor	activation	SRC
Cytokinereceptor	activation	TYK2
Cytokinereceptor	protein–protein interactions	JAK2
Cytokinereceptor	protein–protein interactions	TYK2
ERK1/2	activation	JAK2
ERK1/2	translocation	TYK2
Growthfactor	chemical–protein interactions	Stat3-stat3
Growthfactor	activation	ERK ½
Growthfactor receptor	protein–protein interaction	RAS
Growthfactor receptor	activation	Growthfactor receptor
Growthfactor receptor	activation	Growth factor receptor
Growthfactor receptor	protein–protein interaction	JAK2
JAK2	protein–protein interaction	SRC
JAK2	activation	JAK2
JNK	reaction	SRC
MAP2K1/2	activation	STAT3
MLK	activation	STAT3
MYC	activation	Stat3-Stat3
Mapkkinase	activation	ERK ½
Mapkkinase	activation	Mapkkinase
NDUFA13	activation	CDC25A
P38MAPK	activation	JNK
PIAS3	protein–protein interactions	P38MAPK
PIAS3	activation	STAT3
PIM1	inhibition	Stat3-stat3
PTPN2	protein–protein interactions	Stat3-stat3
PTPN6	activation	Stat3-stat3
RAC1	inhibition	BCL2
RAF1	inhibition	Stat3-stat3
RAS	activation	JAK2
RAS	activation	MLK
SOCS	activation	MAP2K1/2
SRC	activation	RAC1
SRC	inhibition	RAF1
STAT3	activation	JAK2
STAT3	activation	RAS
Stat3-stat3	reaction	STAT3
Stat3-stat3	translocation	Stat3-stat3
Stat3-stat3	activation	STAT3
Stat3-stat3	activation	CDKN1A
Stat3-stat3	activation	MYC
Stat3-stat3	causation	PIM1
TYK2	membership	Transcription

## Data Availability

The data presented in this study are available on request from the corresponding author.

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
