# Peer review of "Effects of S-Adenosylhomocysteine Hydrolase Downregulation on Wnt Signaling Pathway in SW480 Cells"

_ijms, 2023, doi:10.3390/ijms242216102_

Round 1

Reviewer 1 Report

Comments and Suggestions for Authors

The manuscript submitted for publication by Oliver Vugrek and collaborators reports further interesting information regarding the role of S-adenosylhomocysteine hydrolase (AHCY). As it is known that S-adenosylhomocysteine (SAH) hydrolase deficiency is responsible for an autosomal recessive disorder of methionine metabolism, caused by pathogenic variants in the AHCY gene. The manuscript is well described, and the results appear consistent with the conclusions. I have no comments to add

Author Response

We are highly grateful  to this reviewer for the supportive comments to our manuscript.

Reviewer 2 Report

Comments and Suggestions for Authors

S-adenosylhomocysteine hydrolase (AHCY) deficiency causes a potentially lethal rare disease. To investigate its molecular aspects, the authors knocked down AHCY expression in the SW480 colon cancer cell line, mimicking AHCY deficiency conditions. mRNA sequencing revealed differentially expressed networks, notably affecting the LEF1 gene, a key player in the Wnt/β-catenin signaling pathway and stem cell biology. Western blot analysis confirmed elevated LEF1 protein in AHCY-deficient cells, establishing a novel link between AHCY and the cancer cell phenotype. 

This study is at a very preliminary stage in terms of manuscript preparation, data presentation, and overall quality. It lacks a clear rationale for connecting AHCY alteration and CRC (Colorectal Cancer). The majority of the data figures have been generated using IPA (Ingenuity Pathway Analysis). While the English used is not poor, it could be improved for better clarity. Additionally, the available data are insufficient to draw the main conclusions. Below are specific points for the authors to address.

1. Lines 8-9: It seems like two sentences, but there is no period (full stop).

2. Line 9: Needs to be passive (was described).

3. It is unusual to include citations in the Abstract.

4. There is too much experimental detail in the Abstract.

5. Line 46: Perhaps reference 15 is the only source showing the connection of AHCY to cancer. A reference for CRC and AHCY is required. More scientific rationale for this study (AHCY mutation and CRC) should be added; otherwise, the entire study lacks meaning.

6. The authors did not explain why LEF1 was specifically chosen. This reviewer assumes that the authors may want to connect AHCY and CRC, and Wnt is the critical pathway in CRC. This might be the reason they chose LEF1 as one of the upregulated genes in AHCY-KD cells. However, a better rationale for selecting LEF1 is needed.

7. Why are there no Figure 1a and 1b in Figure 1?

8. There are no statistics in Figure 1. P-values are needed to assess the significance of the differences.

9. Figure 2: There are no x and y-axis labels. Additionally, the data may not be essential for the manuscript and could be kept in the laboratory notebook since it doesn't demonstrate any important findings for drawing study conclusions.

10. In the first paragraph of the Results section, the authors presented SAM/SAH measurements. It's unclear what the conclusion is based on these findings. The numbers are compared, but there is no interpretation of what the results mean.

11. Line 109: The authors made a double parentheses error. Similar minor errors were found multiple times, but only a few are mentioned in this review report.

12. Line 127: As mentioned earlier, there is no clear rationale for connecting the study (AHCY) to LEF1. Is it only because a CRC cell line was used?

13. Line 140-141: "Most interesting change" - it's unclear what is meant by the most interesting. Is the highest increase considered interesting?

14. Line 144-145: "Highly increased" - there is no data supporting this statement.

15. Line 166: Why do the authors suddenly focus on MYC and MYC-regulated proteins? Is there any scientific rationale for this?

16. Line 191-197: The first half of this paragraph needs some references to support the statements made.

17. Figure 13: It may not be necessary to include antibody information in the figure legend. It appears that Fig 13 is the only data validating gene expression obtained from RNA-seq and shows only two genes (LEF1 and STAT3). Validation of more gene expressions by western blotting is required.

18. Line 368: "Significantly increased protein levels" - this statement is not accurate because there are no statistics to compare protein levels between the control and AHCY-KD.

19. Why do the authors focus on LEF1? Many other genes may also be increased in KD cells. Is it because CRC cells were used to connect to Wnt signaling?

20. Why do the authors think AHCY-KD activates Wnt signaling? Perhaps this is beyond the scope of their study, but they should at least discuss the potential mechanism of AHCY-KD-mediated Wnt activation.

21. In the Discussion section, there doesn't appear to be a clear focus on genes. Many genes were introduced and discussed, but they have not been validated by western blotting or RT-qPCR. Additionally, it's unclear why MYC was chosen as one of the study's focuses.

22. In the Discussion section, it may not be necessary to discuss many other genes related to all the pathways the authors have analyzed.

23. Lines 591-592: These need to be removed.

24. In the Materials and Methods section, it seems that the Viability Assay and Antibiotic Resistance Testing need to be combined because they need to be conducted as one experiment.

25. Line 796-798: This statement is not correct due to a lack of data. The authors need to test if AHCY overexpression suppresses LEF1, leading to the suppression of all the pathways the authors analyzed.

Comments on the Quality of English Language

The quality of the English language used is of a moderate level. There is ample room for improvement.

Reviewer 3 Report

Comments and Suggestions for Authors

The manuscript explores the impact of S-adenosylhomocysteine hydrolase (AHCY) knockdown in the SW480 cell line. The study reveals significant alterations in gene expression and signaling pathways due to AHCY deficiency, with a particular focus on the role of LEF1 protein. In summary, this manuscript provides a comprehensive understanding of the intricate interplay between AHCY deficiency and multiple signaling pathway. However, key questions remain:

1. The authors should perform quantitative PCR (qPCR) or other validation techniques to confirm the observed gene expression changes identified through RNA-seq analysis.

2. Besides LEF1 and STAT3, the authors should also verify the changes in additional protein expression levels, especially for key proteins identified in IPA analysis, using Western blotting.

3. The authors should conduct functional assays to assess the impact of AHCY deficiency and the identified gene expression changes on cellular processes such as proliferation, migration, invasion, and apoptosis,

4. The authors could investigate the sensitivity of AHCY-deficient cells to certain drugs or compounds targeting the identified pathways in IPA analysis.

Minor Issues:

1) Figure 1a & b are absent.

2) Y-axis labels for figures are not provided.

3) Figures lack statistical analysis, and p-values are not displayed.

4) Clarify the meaning of "2+4" and "SCR" in Figure 13.

5) The Discussion section lacks content regarding Additional Pathways Perturbations.

6) Content is missing from Line 591 "Top of Form" and Line 592 "Bottom of Form".

Round 2

Reviewer 2 Report

Comments and Suggestions for Authors

Line 8: I don't think "AHCY deficiency" is the name of a disease.

-#6: The authors still did not clearly answered why LEF1 is selected.

-#9: The authors simply deleted the data (Figure 2). Need to rename the whole data.(Figure 2 is missing).

-#20: I asked how AHCY-KD increases Wnt activation but the authors stated something totally irrelevant (how Wnt signaling is activated, which is what your can easily find in Googgle.)

-#25: This reviewer cannot find the data (a test if AHCY overexpression suppresses LEF1)

Comments on the Quality of English Language

need some revision.

Reviewer 3 Report

Comments and Suggestions for Authors

The authors have addressed my concerns and I have no further questions.

Author Response

(The authors gave the same response as above.)

Round 3

Reviewer 2 Report

Comments and Suggestions for Authors

1. In Line 8, the authors simply converted AHCY to S-adenosylhomocysteine hydrolase. It's important to note that "S-adenosylhomocysteine hydrolase deficiency" is not the name of a disease, either.

2. I do not fully agree with the rationale provided by the authors, but I appreciate their logical approach.

3. I recommend the removal of Figure 2, as it may not be entirely appropriate for the research manuscript. If removed, please update the Figure numbers accordingly, for example, renaming Figure 3 to Figure 2, Figure 4 to Figure 3, and so on.

4. It would be beneficial if the authors could incorporate a discussion on the potential mechanisms, such as epigenetic changes induced by AHCY-DK, affecting Wnt signaling.

5. The absence of a gain-of-function study (involving the overexpression of AHCY) weakens the overall conclusion.

Comments on the Quality of English Language

No comments

Round 4

Reviewer 2 Report

Comments and Suggestions for Authors

I don't have any further comments on the authors' response. One last point is as follows: In Figure 12, the figure legend needs to be placed below the figure, not above it.